# Active Video Games for Improving Mental Health and Physical Fitness—An Alternative for Children and Adolescents during Social Isolation: An Overview

**DOI:** 10.3390/ijerph18041641

**Published:** 2021-02-09

**Authors:** Isis Kelly dos Santos, Rafaela Catherine da Silva Cunha de Medeiros, Jason Azevedo de Medeiros, Paulo Francisco de Almeida-Neto, Dianne Cristina Souza de Sena, Ricardo Ney Cobucci, Ricardo Santos Oliveira, Breno Guilherme de Araújo Tinoco Cabral, Paulo Moreira Silva Dantas

**Affiliations:** 1Graduate Program in Health Sciences, Federal University of Rio Grande do Norte, Natal 59078-970, Brazil; isisk2@ufrn.edu.br (I.K.d.S.); rafaelacath@hotmail.com (R.C.d.S.C.d.M.); jason.medeiros1@hotmail.com (J.A.d.M.); 2Graduate Program in Physical Education, Federal University of Rio Grande do Norte, Natal 59078-970, Brazil; paulo220911@hotmail.com (P.F.d.A.-N.); diannesena@hotmail.com (D.C.S.d.S.); 3Biotechnology Graduate Program, Potiguar University of Rio Grande do Norte, Natal 59078-970, Brazil; rncobucci@hotmail.com; 4Department of Physical Activity, Federal University of Rio Grande do Norte, Natal 59078-970, Brazil; roliveira.ufrn@gmail.com (R.S.O.); brenotcabral@gmail.com (B.G.d.A.T.C.)

**Keywords:** physical activity, adolescents, health promotion, video games, overviewéó

## Abstract

The aim of this study was to synthesize the evidence on the effects of active video games (AVGs) on mental health, physical fitness and body composition of children and adolescents. A search was conducted in the following databases: PubMed; MEDLINE (by Ovid); SportDiscus, Cochrane library systematic reviews (CENTRAL) and EMBASE with no language restrictions during October 2020. Reviews on the use of AVGs were included in the study. We use the AMSTAR (A MeaSurement Tool to Assess systematic Reviews) scale to analyze the methodological quality of the studies. Seventeen systematic reviews and meta-analyzes were included on the effects of AVGs with 30 to 4728 children and adolescents of both sexes with ages ranging from 6 to 19 years. In five studies, the population was overweight or obese. Regarding the quality, 12 studies were of moderate quality, two had high quality, two had low quality and one showed very low quality. The analyzed data indicate that the use of AVGs with a frequency of 1 to 3 times a week with durations of between 10 and 90 min per day shows positive effects on mental health and physical functioning. There was moderate quality evidence that AVGs can result in benefits for self-esteem, increased energy expenditure, physical activity and reduced body mass index in children and adolescents who used AVGs in the home environment. Further research is needed on this tool to help in the process of social isolation and consequently in promoting health and well-being.

## 1. Background

The emergence and rapid spread of coronavirus disease (Covid-19), a disease that promotes severe acute respiratory syndrome caused by the new coronavirus (SARS-Cov-2), motivated the World Health Organization (2020) to determine a state of emergency worldwide, characterizing it as a pandemic [1,2,3]. In order to delay the spread of the virus, social distancing is currently recommended. As a consequence, the United Nations, Education Organization and the Scientific and Cultural Organization suggested closing schools and universities in 190 countries worldwide [4].

In view of this scenario, schools and universities were forced to suspend classes indefinitely, causing children and adolescents to spend most of the time at home, or with very restricted coexistence [5,6]. These restrictions became a challenge for everyone due to the reorganization of school activities moving onto the online format, and the need to organize and maintain a systematic routine for the family so as not to disrupt leisure and physical activities [7,8]. As a result, long periods of social distancing may have increases in anxiety, stress and change of physical activity routine of children and adolescents [9,10].

In a retrospective study De Matos et al. [11] have recently shown in a sample of Brazilians that, in the period before the COVID-19 pandemic, children and adolescents of both sexes had a significant higher level of physical activity and energy expenditure, and were lower in body weight compared to the period of self-distancing imposed by COVID-19 [11].In addition, the authors emphasize that during social distancing the domains of quality of life (e.g., functional capacity, general health status, vitality, social emotional aspects and mental health) decrease significantly in children and adolescents [11]. 

According to Galvin et al., [12] and Ramchandani et al., [13], approximately 4.4 million children aged 3 to 17 years were diagnosed with anxiety and depression due to isolation during the COVID-19 pandemic. This information converges with data from the Center for Disease Control and Prevention (CDC) [12]. In addition, studies conducted since the beginning of social distancing have reported that children and adolescents are experiencing a significant increase in mental health disorders classified as mild to severe, such as anxiety, depression and sleep disorders [14,15].

The aforementioned facts occurred because children and adolescents were prevented from practicing physical activity and from having moments of leisure. In fact, evidence suggest that involvement in regular physical activity is positively associated with increases in self-esteem and prevention of mental problems [15]. Therefore, there is a need to investigate new strategies that can prevent damage to the health of children and adolescents [15], especially considering that health professionals need to develop efficient strategies for the improvement of mental and physical health during the social isolation resulting from COVID-19 [16].

In the last few decades, young people have spent a long time doing sedentary activities such as watching television, using their cell phones, and sedentary video games. Studies suggest that the use of active video games (AVGs), including electronic motion games, contribute to increasing daily caloric expenditure, interaction and social support, and can be used as strategies for coping with the low level of physical activity of young people during the pandemic [17,18,19]. Thus, studies with the use of interactive video games have begun to be conducted in order to use technology to “break” from the sedentary lifestyle [20,21,22,23] by performing physical activities with light to moderate intensity, encouraging and increasing energy expenditure [24,25]. In this way, in a context of social distancing, which requires commitment to take care of one’s health with adequate daily habits, active games can be an interesting alternative to provide greater mental health, increase physical activity and family interaction. 

Considering that AVG’s are attractive to children and adolescents, it seems feasible to use AVGs as a strategy for the practice of physical activity in this population [14,17,23]. Furthermore, the increase in physical activity has been positively associated with mental health [24]. Candidate mechanisms are the release of endorphins, dopamine, serotonin and oxytocin that promote neurochemical reactions that amplify the sensation of pleasure of subjects, thus promoting the reduction of depressive symptoms and the improvement of mental health [26]. Thus, research on the effects of AVGs can expand the knowledge about this strategy as a means to combat the losses in mental health caused by social isolation [16]. 

Therefore, the present study aimed to review scientific evidence on the effects of AVGs on mental health (self-esteem) and physical functioning (energy expenditure, physical activity and body composition) of children and adolescents. 

## 2. Methods

We conducted this overview and registered the review on PROSPERO (CRD42020181767), providing updates to the protocol. We followed the guidelines of the Preferred Reporting Items for Systematic Reviews and Meta-Analyses (PRISMA) statement [27]. We included systematic reviews and/or meta-analysis about the benefits of AVGs on mental health and physical activity in children and adolescents (3–19 years) without neurological conditions or neuromuscular diseases. Were included in the overview articles published from 2010 to 2020. We considered interventions that studied active games played on the most commonly available commercial consoles. We excluded randomized controlled trial (RCT), cohort, case-control and cross-sectional studies.

### 2.1. Electronic Databases and Information

We searched the following electronic databases: PubMed; MEDLINE (*by Ovid*); SportDiscus, Cochrane library systematic reviews (CENTRAL) and EMBASE, with no language restrictions. We developed search strategies using a combination of Medical Subject Headings (MeSH) terms and word keys: (child* OR adolescent* OR teenage* OR youth) AND (active video game OR exergame OR interactive game OR health game) AND (“self-esteem” OR “mental health” OR “depression*” OR “anxiety” OR psychological OR psychiatric OR insomnia OR “mental health” OR “well-being” OR “wellbeing” OR “psychosocial”) AND (physical activity OR exercise OR fitness OR energy expenditure OR energy cost) AND (systematic review OR meta-analysis OR scoping review OR integrative review). The last search was conducted in October 2020.

All articles were screened by IKS and RCSCM for title and abstract based on inclusion and exclusion criteria. In cases of disagreement about eligibility, another reviewer (PFAN) helped reach a consensus. Two authors (IKS and RCSCM) screened all full text publications independently. The team conducted literature screening using Ryyan CQRI [28], and one author checked the reference list (JAM). 

### 2.2. Data Extraction 

Data extraction was conducted by two reviewers (PFA-N and DCSS) and verified by a third (RCSCM) for the following elements: study design [including methods, location, sites, groups]; participant characteristics; intervention characteristics; comparator characteristics; and outcomes assessed (numerical data). The following data were extracted from all the eligible articles: (a) author/year; (b) population/condition; (c) age (years); (d) intervention; (e) control; (f) inclusion criteria; (g) exclusion criteria; (h) results of mental aspects; (i) results of physical aspects, and (j) results of social aspects.

### 2.3. Assessment of Methodological Quality of Included Reviews 

Two authors (IKS and RCSCM) assessed the methodological quality of the included systematic reviews using the tool A Measurement Tool to Assess Systematic Reviews (AMSTAR 2) [29]. Discrepancies were resolved by a third reviewer (RNC). AMSTAR is used as a practical critical appraisal tool to conduct rapid and reproducible assessments of the quality of systematic reviews of randomized controlled trials and non-randomized studies. The overall confidence in the systematic review results is rated as: high, moderate, low and critically low.

### 2.4. Data Management

The data were described in a narrative form and synthesized in tables. The evaluated outcomes were self-esteem (mental health), increased energy expenditure, physical activity and body composition (physical functioning). The results were presented using the crude measures pointed out in the studies.

## 3. Results

A flow chart with the number of articles in each step of the review is presented in Figure 1. The initial search strategy retrieved 917 articles, from which 47 were duplicates. Titles and abstracts of 870 studies were screened and 826 were excluded. Full texts of 44 studies were screened and 27 studies were excluded. A total of 17 systematic reviews met the inclusion criteria. The characteristics of the included reviews are shown in Table 1. The excluded studies and their reasons are presented in the Appendix A.

### 3.1. Characteristics of Systematic Reviews or Meta-Analysis

Seventeen studies were included [18,30,31,32,33,34,35,36,37,38,39,40,41,42,43,44,45], from which 7 were systematic reviews and meta-analysis [18,32,33,36,37,44,45]. All studies focused on children and adolescents of both sexes with an age ranged from 3 to 19 years. Sample size ranged from 30 to 4728 participants. In five studies, the population was overweight or obese. The number of studies included in the systematic reviews and meta-analysis ranged from 5 to 52 studies (Table 1). Interventions were conducted between periods of 1 week to 9 months, with a duration of between 10 and 90 minutes per day and frequency of active exercises at home from one to three times per week.

Among the 17 studies, four presented data regarding the effect of playing AVGs improve on self-esteem, self-efficacy and socialization, ten reported results of the participants’ physical health, such as increased energy expenditure and physical activity while playing AVGs. Finally, four studies presented relevant data on the reduction of some parameters of body composition after physical activity from playing AVGs. No systematic reviews were carried out with studies during the pandemic. However, all inserted studies were carried out in the home context.

#### Methodological Quality Assessment

Most of the systematic reviews and/or meta-analyzes included were of moderate quality (12 studies), that is, although there are some methodological flaws, these studies are able to accurately summarize the results. Two studies with high quality [44,45], two studies with low quality [34,39] and one study with critically low quality [31] were included in the review. The results of the analyzes are shown in Table 2. The weaknesses observed in the included studies are: lack of information about registration of the protocol, lack of exhaustive strategies search, lack of justification for studies exclusion, and lack of satisfactory explanation of the heterogeneity observed in the results.

### 3.2. Types of Exergames

The included studies investigated the effect of the following activities: dance, boxing, walking, running, jumping and bowling [37,44]. The AVGs used were: Wii Balance, Wii Aerobics, Wii Boxing, Wii Golf, mini basketball indoor video game version and dancing to music [30,36,39]. The AVG type has been coded based on the main body movements needed to play: upper body movements (e.g., Wii sports), lower body movements (e.g., Dance Dance Revolution) or full body movements (e.g., PS2 Eyetoy Final Furlong). On the contrary, in the study by Peng et al. [36], multiple AVGs and physical activities or data reported at various times were addressed, for example, brisk walking three times for three types of AVGs (Wii tennis, Wii baseball and Wii boxing). When investigating the types of consoles used to practice exergames in the included studies, it was possible to observe that the Nintendo Wii, Dance Dance Revolution, Exerbike XG, Makoto Interactive Arena, Treadwall, Xavix, PlayStation Eye Toy, Microsoft Xbox 360 (Microsoft Inc, Redmond, WA, USA) and game controllers such as CatEye Gamebike + Sony GameCycle (Sony Computer Entertainment Inc, San Mateo, CA, USA), Sony PlayStation 2 (EyeToyVR) (Sony Computer Entertainment Inc, San Mateo, CA, USA), PlayStation 2(Sony Computer Entertainment Inc, San Mateo, CA, USA) and Kinect^TM^ were considered eligible. Commercially available active video games have different ways of playing and intensities; most of them with remote controls (Nintendo Wii, Nintendo, Kyoto, Japan) or PlayStation Move (Sony Computer Entertainment Inc, San Mateo, CA, USA) or through the user interface using gestures captured by a webcam (XBOX 360 plus Microsoft Inc, Redmond, WA, USA). To analyze data on activity measurement some studies used accelerometers, Fitbits, pedometers, and self-reports (questionnaires validated with the Perceived Competence Scale) [43].

### 3.3. Effects of Exergames

The findings are summarized below and presented in detail in Table 2.

**Table 2 ijerph-18-01641-t002:** Summary of results and methodological quality of included studies (n = 17).

Outcomes				
Authors, Years	Mental Health	Physical Health	BMI	Quality of the Evidence ¹
Andrade et al., 2019	←→ self-esteem and self-efficacy			High
Ameryoun et al., 2018			↓BMI	Moderate
Barnett et al., 2011		↑ level of energy expenditure		Critically Low
Bochner et al., 2015			←→ weight change	Moderate
Carmo et al., 2012		↑ physical activity		Moderate
Costa et al., 2020	↑socialization			Low
Gao & Chen, 2014		↑ physical activity	←→ BMI	Moderate
Gao et al., 2015	↑ self-esteem	↑ level of energy expenditure ↑ physical activity		Moderate
Jimenéz et al., 2019			↓BMI	Moderate
Joronen et al., 2017	↑ self-esteem and self-efficacy			Moderate
Lamboglia et al., 2013		↑ level of energy expenditure; maximal oxygen uptake; heart rate↑ physical activity		Low
LeBlacn et al., 2013		↑ level of energy expenditure		Moderate
Lu et al., 2013			↓BMI	Moderate
Oliveira et al., 2020		↑ physical activity	↓BMI	High
Pakarinen et al., 2017		↑ physical activity		Moderate
Peng et al., 2011		↑ level of energy expenditure; maximal oxygen uptake; heart rate		Moderate
Williams & Ayres, 2020		↑ physical activity		Moderate

Note: ↑ = increase; ↓ = decrease; ←→ = uncertainty; Quality of the evidence ^1^ = based AMSTAR 2.

#### 3.3.1. Effects of Mental Health 

Four systematic reviews analyzed the effect of AVGs on mental health of children and adolescents [18,39,40,45]. One study synthesized the effects of AVGs on self-esteem of children/adolescents [18]; another investigated the effectiveness of educational technologies for promoting socialization [39]; another reported the effects of exergames on well-being [45]; and one analyzed the psychological effects (self-esteem and self-efficacy) of children and adolescents who were overweight or with obesity [40]. It is noteworthy that the comparisons between studies led to divergent results in different situations. For example, when comparing the intervention of AVGs with sedentary behaviors (resting, sitting, playing sedentary video games and watching TV), laboratory exercises (walking and cycling) and physical activities based in the field (recreation and aerobic dance). 

Methodological quality ranging from high to low suggested that interventions with AVGs can improve the self-esteem, self-efficacy (perception of success and confidence) and socialization of children and adolescents when compared to a control group (children not performing any type of activity) [18,39,40,45]. In addition, children prefer practices with exergames over other activities (e.g., walking on the treadmill). However, most comparisons produced small effect sizes (ES). When comparing overweight, obese or children and adolescents with other conditions, it was found that they feel more satisfied when practicing exergames than those with normal weight, showing greater self-efficacy, positive expectations and satisfaction with AVGs [18,40]. Furthermore, AVGs presented positive results in motivation, pleasure, and psychological and social well-being; in addition to offering different learning experiences [39,45]. Finally, it is suggested that they can be affective in improving psychological aspects, such as physical engagement [40]. 

No study reported negative results from active/interactive exergames on mental health in children and adolescents. On the contrary, when investigating the use of AVGs in competition mode, no difference was found in the attraction of the game and intrinsic motivation, however, in the cooperation mode, a positive effect on mental health was found [45].

#### 3.3.2. Effects on Physical Fitness

Ten systematic reviews presented data on physical fitness, here defined as maximal oxygen uptake, heart rate, energy intake, and increased physical activity of children/adolescents [18,30,31,34,36,38,41,42,43,44]. Seven reviews of predominantly low to moderate quality showed the effects of AVGs on some aspects of physical functioning [18,30,36,38,41,42,43]. The responses induced by AVGs are greater when compared to children and adolescents who demonstrate sedentary behaviors, showing substantial effects on energy expenditure, on the heart rate identified during the intervention with exergames, on the rate of perceived effort, and on the maximum consumption of oxygen (VO2 max) [18,30,34,38,42,43,44]. In addition, seven reviews [18,30,34,38,42,43,44] analyzed the effects of exergames on physical activity levels. Evaluation of the quality of the reviews ranged from moderate to high and the results showed positive effects on the promotion of increased self-reported physical activity, in addition to greater adherence by the pediatric public. Moreover, after being involved with active exergames, there was a reduction in sedentary behaviors [34,38]. However, because some studies have small samples and a high dropout rate, the results should be interpreted with caution. 

Active exergames are associated with increases in acute energy expenditure, with the potential to increase levels of physical activity, but the effects on habitual physical activity are not clear [41,42]. Costa et al., [39] add that positive results were also achieved in physical activity, but the results are more satisfactory when using healthy lifestyle strategies (diet + activity). Regarding the child development process, one result suggested aspects related to constructivism and organic maturation [39].

#### 3.3.3. Effects on Body Composition

None of the included reviews reported responses on the effects of AVGs on body composition. However, six reviews showed results of AVGs on the body mass index (BMI) of children and adolescents [30,32,33,35,37,44]. In four moderate quality reviews, AVGs were identified as having a positive impact on weight reduction and BMI [30,33,35,37]. On the contrary, two reviews did not find a significant difference between the intervention and control groups in terms of reduction in BMI [30,32], revealing that interventions based on games have a relatively small effect on improving BMI in obese or overweight children [32].

## 4. Discussion

We investigated the effect of active video games (AVGs) on the mental health and physical fitness of children and adolescents. Our findings demonstrate divergent results on the impact of AVGs on mental health, however, some benefits are promoted by exergames from the perspectives of self-esteem and socialization. The physical fitness results showed improvements in energy expenditure, oxygen consumption, heart rate and physical activity level in children and adolescents. This is the first overview investigating the effect of active video games (AVGs) and there are some divergent findings due to the different designs or heterogeneity in the systematic reviews and meta-analyzes included. 

The divergences found in the present study can be explained by the fact that exergames are a relatively new technological achievement, making studies with different designs possible. However, it should be understood that promising opportunities exist with technologies that, for example, make people more active and promote motor skills, cognitive performance and mental health [46,47]. In terms of mental health, it is noteworthy that the present study found that exergames promote positive effects on self-esteem, self-efficacy, self-concept, help with interest and situational motivations, providing a sense of pleasure, psychological well-being and different learning experiences [30,41,43]. 

Exergames promote increased self-efficacy, intrinsic motivation, prosocial behaviors and continuous gameplay. However, it is noteworthy that there is a difference with people who play cooperatively in the form of competition, as those who are highly competitive report more positive moods and greater pleasure when placed in a competitive exergaming environment, while non-competitive participants report higher levels of pleasure, humor, self-efficacy, motivation, engagement and play for longer periods when placed in a non-competitive exergaming environment. Thus, video games can be beneficial to the brain, however, the benefits vary depending on the type of video game [47,48,49]. 

It is noteworthy that in the pandemic period, children feel their parents’ stress and can demonstrate their concerns in different ways by becoming bored, idle, stressed, agitated and having sleeping disorders. In view of this concern, different strategies, interventions and/or resources available at home should be used in the routine to minimize these feelings promoted by social isolation and help with the child’s well-being, such as the use of active AVGs that are based on the idea of integrating physical activity and exercise in digital games and have the potential to positively impact cognitive and psychosocial variables [47,48]. During large-scale pandemics and disasters, to minimize the impacts on the mental health of the general population and especially the pediatric population, family leisure activities and games involving physical activities such as AVGs are recommended [49]. In this sense, a systematic review pointed out that adaptation of the home environment to perform physical activity and/or exercise during the quarantine period contributes to the improvement of mental health through psychological well-being [50].

Among the physical fitness outcomes, the results were divergent due to different comparisons, types of AVGs, and players’ ages. However, most of the included studies have shown that AVGs are capable of increasing energy expenditure, oxygen uptake, and heart rate during the game [18,41]. Regarding energy expenditure, it was observed that AVGs do not reach vigorous intensity, however, the findings suggest that AVGs can be recognized as tools or alternatives for reducing sedentary behavior, adding to physical activity (from light intensity to moderate). The levels of physical activity during the game are increased, and may even provide greater energy expenditure in the pediatric population [51]. AVGs lead to physical activity of mild to moderate intensity, actually improving the level of physical activity [24] and motor skills [20]. This can be explained by the fact that exergames actively provide feedback on the amount of time spent, which increases awareness and self-regulation, encouraging and helping young people to maintain healthy behavior [52], mainly due to the potential for individualization, adaptability and specificity [31]. 

During the pandemic, there is an early concern about sedentary lifestyle and obesity, in addition, studies conducted within the COVID-19 pandemic observed an increase in the z-scores of body mass index and an increase in the prevalence of childhood obesity [53]. The impact of COVID-19 on childhood obesity was modestly greater among boys and people of color, and health interventions are needed to promote an active lifestyle and involve children in physical activities in order to minimize the impact of the pandemic on weight gain and childhood obesity [53]. Thus, it is essential to maintain the involvement of active behaviors at moderate and controlled levels, especially during the pandemic [54]. Cooperative and/or competitive AVGs can produce weight loss in overweight and obese children and adolescents [49]. Exergames are attractive to children due to the fun offered and/or motivating resources, but they promise to be an ideal intervention only if they replace sedentary activities (sedentary video games/watching TV) [19]. 

Systematic reviews have shown that AVGs have a positive impact on the social life of children and adolescents, as active games have a recreational and socialization approach [39]. Studies claim that some exergames offer a unique and exciting opportunity for positive effects on social interaction and cooperation in this audience [17,21]. Specifically, in adolescence, it is known that an important issue is social relationships and what makes this aspect very difficult is the need for isolation at home, causing frustration, nervousness, discontent, nostalgia and boredom because of social distancing [48].

In this sense, active games that involve cooperation and competition can minimize these aspects. 

On the contrary, negative or antisocial disorders may develop in children due to the excessive use of electronic media [52]; however, if family members monitor and regulate children’s behavior by developing rules and participating in activities, this problem can be avoided. It is known that if parents themselves are role models, regulating their own technology-related behaviors, this can also help their children establish controlled use [55]. In addition, parents should encourage the active participation of their children, playing video games with them, since the involvement will help to regulate the use and to know the characteristics of the different video games, in addition to promoting adaptable online activities, reducing the use of a “sedentary” screen for all [52]. 

Analyzing critically, all aspects and substantial modifications promoted by exergames should be monitored continuously to avoid potential threats, especially considering the risk of replacing traditional physical exercise along with the risk of increased screen time [56]. In contrast, it is believed that the active behavior promoted by AVGs will help in the daily lives of many families who are isolated at home, under great stress. A study points out that exergames can be useful in the formation and maintenance of meaningful relationships, which can help to reduce feelings of loneliness during physical distancing and improve the quality of life for everyone [52]. 

### 4.1. Practical Implications

The results shown in this review support the benefits of using AVGs in reducing physical inactivity and in the improvement of emotional and cardiovascular aspects that are important for the quality of life of children and adolescents. There is evidence to suggest that AVGs have the potential to increase habitual physical activity among children and young people compared to sedentary video games [38,40]. However, it is important to explore the use of AVGs as a strategy to reduce physical inactivity, and subsequently examine the impact of their use on a series of health, behavior and time-of-use indicators, as the literature points out and reinforces that excessive screen use is associated with harmful health effects [55]. Moreover, the recommendation of physical activity for children and young people is 60 minutes or more daily, involving a variety of activities, and in this sense, using AVGs encourages the reduction of a sedentary lifestyle and is more desirable when compared to the use of just a sedentary screen [56,57]. Although it is important to mention that it is not desirable that AVG play is comprised of the total physical activity recommended for this age group, but yes, it could constitute an important part of this requirement.

### 4.2. Limitations

Some limitations may limit the interpretation of the results. For example, most of the included studies were conducted before the pandemic. Moreover, the majority presented moderate quality and others presented low and critically low quality. In addition, differences in the samples and types of AVGs evaluated in the studies may have influenced the results and caused divergences regarding the benefits of physical and mental health, this review, given the short period of time to which refers, have little bibliography to obtain strong conclusions and similar studies will need to be carried out with more published works.

## 5. Conclusions

Most studies included in this review with moderate quality indicated that active video games improved any aspects of health mental (self-esteem), physical activity and energy expenditure in children and adolescents, since exergames offer a unique and exciting opportunity for positive effects on social interaction and cooperation in this audience. However, the results need to be analyzed with caution, since divergences exist among systematic reviews and these studies were conducted before the pandemic period. New studies should be completed aiming to evaluate the effects of the AVGs on mental and physical health of children and adolescents during this period of social isolation. This would increase our knowledge about the use of this tool to reduce the negative impacts caused by COVID-19.

## Figures and Tables

**Figure 1 ijerph-18-01641-f001:**
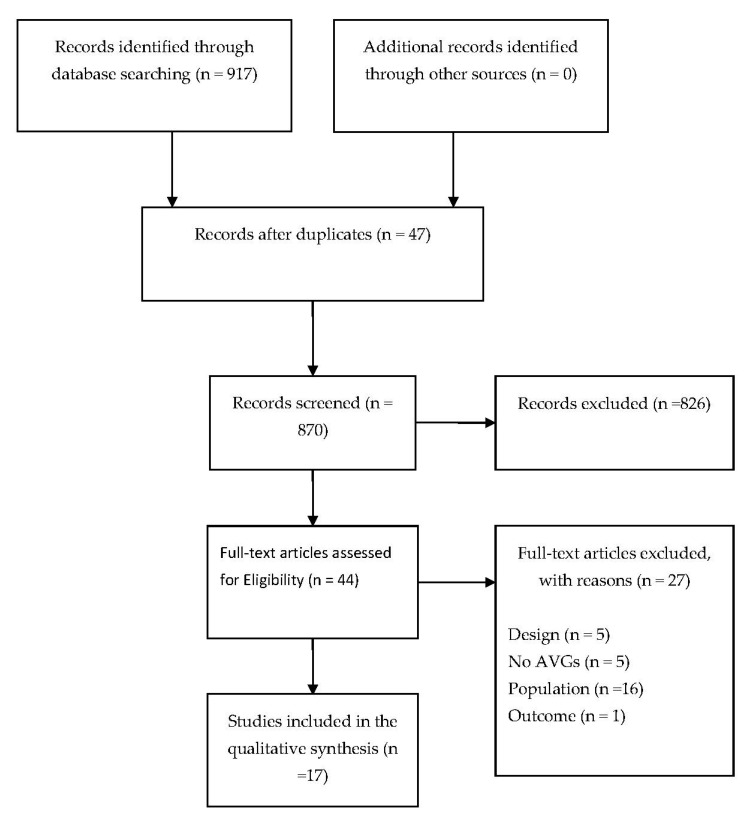
Study Flow diagram.

**Table 1 ijerph-18-01641-t001:** Characteristics of systematic reviews or meta-analysis.

Authors, Years	Study Design	Country	Number of Trials	Participants (Number)	Gender Distribution	Participants Ages (Mean-Years)	Population/Diagnoses	Intervention	Control
Ameryoun et al., 2018	Systematic review and meta-analysis	Iran	11	951	Both sex	5–18	Overweight or obese	Active game (based in: soccer, dance, cycling)	No Control
Andrade et al., 2019	Systematic review and meta-analysis	Brazil	9	336	Both sex	6–19	Overweight or obese	Exergames	walking on the treadmill, control
Barnett et al., 2011	Systematic review	China	9	187	Both sex	6–18	Overweight and non-overweight	AVGs	Walking, running, squatting, jumping, stamping, dance, boxing;
Bochner et al., 2014	Meta-analyses	USA	7	588	Both sex	7–19	Overweight or obese	AVGs	no-intervention
Carmo et al., 2012	Systematic review	Brazil	10	623	Both sex	8–14	No related	AVGs	No intervention
Costa et al., 2020	Systematic review	Brazil	8	4728	Both sex	3–18	No related	Educational technology	Controlled and Uncontrolled
Gao & Chen, 2014	Systematic review	USA	34		Both sex	7–19	Obesity	AVGs	
Gao et al., 2015	Meta-analysis	USA.	35	11 to 1112	Both sex	6–15	No related	AVGs	Laboratory based exercise exercises, sedentary behaviors
Jimenéz et al., 2019	Systematic review and meta-analysis	Spain	16	1222	Both sex	7–18	Overweight or obese	Active game (based in: game bike, cycling, dance met)	Control group
Joronen et al., 2017	Systematic review	Finland	10	882	Both sex	9–19	Normal weight; overweight or obese	AVGs	No-exercise control group
Lamboglia et al., 2013	Systematic review	Brazil	9	516	Both sex	6–15	Obesity	AVGs	Conventional sedentary games/No intervention
LeBlacn et al., 2013	Systematic review	USA	52	1992	Both sex	3–17	Overweight and obese	AVGs	Passive video game/No intervention
Lu et al., 2013	Systematic review	Illinois	14	1349	Both sex	7–18	Overweight and obese	Games	No intervention
Oliveira et al., 2020	Systematic review and meta-analysis	Brazil	12	1016	Both sex	7–19	No related	AVGs	Normal actives or game passives
Pakarinen et al., 2017	Systematic review	Finland	5	726	Both sex	9–19	No related	Exergame (Dance Revolution, interactive multimedia for improving FITness)	No intervention
Peng et al., 2011	Meta-analysis	Michigan	18	354	Both sex	<18–>18	Normal weight and Overweight	AVGs	while engaging in other traditional physical activities
Williams & Ayres, 2020	Systematic review	USA	6	20 to 105	Both sex	12–19	Overweight and obese	AVGs	Control condition/received a Fitbit

## Data Availability

The data that support the findings of this study are available from the corresponding author, P.M.S.D, upon reasonable request.

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
