# Peer review of "Active Video Games for Improving Mental Health and Physical Fitness—An Alternative for Children and Adolescents during Social Isolation: An Overview"

_ijerph, 2021, doi:10.3390/ijerph18041641_

Round 1
Reviewer 1 Report
In the last years, active video games have become a form of leisure activities and an important element of rehabilitation. They are an attractive form - especially for children.
This review assumes to show the importance of active video games for physical and mental health. This topic seems to be particularly important because of the pandemic, which the authors mention.
However, the authors should better describe the impact of active video games on mental and physical health, what changes have occurred and how many reviews. It is important to have a proper reference to activities in social isolation, whether and how the game can improve psychophysical condition.
The study is properly prepared, innovative.
The subject matter is current, extremely important.
Any social isolation can contribute to psychophysical disorders, this problem also concerns children and adolescents.
During social isolation, due to the pandemic , children and adolescents were prevented from practicing physical activity. Physical activity may play an important role in the prevention of mental health diseases, especially depression and anxiety.
It is worth and necessary to look for new strategies that can prevent damage to the health of this population. Mental diseases are a serious public health issue. Depression and anxiety are some of the conditions that affect young people at disproportionate rates in comparison to many other population groups.
Active video games can be promising. All inserted studies were carried out in the home context. They can be referred to as the time of isolation.
However, the authors should better describe the impact of active video games on mental and physical health, what changes have occurred and how many reviews.
The part following requires a more detailed description
3.3.1. Effects of mental health
3.3.2. Effects on physical functioning
for example:
227: "the effect of AVGs on the mental health" - what was examined
245: "the effects of AVGs on some aspects of physical functioning" - what was examined
Authors should add information about limitations.
Author Response
Dear Reviewer, we thank you for the excellent comments and for the opportunity to revise the manuscript, improving its quality and bringing it within the standards required by IJERPH.
Thank you for your revision. The sections 3.3.1, 3.3.2 and limitations were adjusted to include the reviewer's suggestion.
Reviewer 2 Report
I believe that the work is well done but there are very few studies that have met all the criteria and, furthermore, the works that have been finally suitable are not very good or good works and the results are not suitable for drawing conclusions with much base. Therefore, although the subject is interesting, we should wait longer to do this study since we have not really been with the virus for a year and therefore I think I have been hasty to draw serious conclusions. That is why I have requested that the article be rejected.
Author Response
Dear Reviewer, we appreciate the reviewer's comments. We conducted the review in order to instigate the debate about the effects of the pandemic on children and adolescents' health. Moreover, we also addressed possible solutions to improve the quality of life of this population during the period of social distancing caused by the COVID-19 pandemic. In the last year, studies showed a decrease in physical activity and health markers of the youth population.
Reviewer 3 Report
my congratulation for this review. overall, for the time efficient production.
I suggest a little improvement in conclusion section (line 365): indicate between brackets the way or rationale as the exergames/videogames improve the self-esteem.
Author Response
We appreciate your comments. The conclusion was adjusted and now includes your suggestion.

Reviewer 4 Report
I have carefully reviewed the manuscript entitled "Active video games for improving mental health and 3 physical functioning - an alternative for children and 4 adolescents during social isolation: an overview". This umbrella review summarizes the evidence of 17 systematic reviews and meta-analyzes examining the relationship between AVGs, mental health and physical functioning in school-age children and adolescents. First of all, I must say that I am not an expert in systematic reviews. However, it seems that the authors were rigorous with the method and the results are well presented. As a counterpart, I found the introduction section a bit vague in some key points (e.g., mechanisms to explain the association between AVGs and outcomes, GAPS, prior evidence). Likewise, the discussion section should be highly reinforced with explanations and prior research arguments. These were my major concerns and both should be addressed. Furthermore, it seems necessary for the authors to reflect on what is the threshold of the benefits of AVGs on the health of children. What are the practical implications? Always more AVGs is better? Or can abusive screen use have parallel harmful effects? These points should be addressed in the discussion. Finally, I am also not a native speaker and my English is not perfect, but I have still detected numerous spelling errors inappropriate for a quality scientific study. Also, with typographical errors that must be reviewed. These minor details, though less, also need to be addressed. See below:
Abstract:
The abstract is well written and makes the paper easy to understand. However, the following details should be reviewed:
1) In the abstract and throughout the manuscript, numbers less than 10 should be written with letters instead of Arabic numbers (ie, L24, 2 had high ... 2 had low ... 1 showed ...; L26, of 1 to 3 ..., among others; L 178…180, 182….)
2) Please, here, and throughout the manuscript, review the punctuation marks. Avoid using sentences that are too long. For example, in L30, there is a ";" which should be replaced by a ".".
3) L28. The term active video games should be abbreviated, or always, or never. Please check.
Introduction:
4) Although the introduction seems well written (a bit short and vague as well), there are some sentences that are not well understood. For example, between L45 and L49 authors wrote: “Long periods of social isolation can impact mental health causing increased anxiety, and the entertainment and physical activity routine of children and adolescents are affected by losses relating to the lack of social coexistence and exercises, these include low self-esteem, being overweight or even obese”. In my view, is a sentence too long that could be split to improve its understanding. In addition, authors should be more accurate with references because in this sentence, for instance, there are different findings that could be referenced (i.e., linked) with its specific study. Please, check.
5) Please, check for typographical errors throughout the text (e.g., L 51).
6) L61 to 63. This is another example where the authors need to be more precise with the references.
7) L71-73. The objective of the research is interesting, even more so in these times of pandemic. However, the introduction is a bit vague up to this point. I think the authors misidentify GAP. Nor do they detail the practical contributions that their review could have on this subject. These points should be reinforced in the introduction. In addition, the authors should delve deeper into the state of the art on the effects of AVGs in children outcomes.
8) L.72. AVGs have already been abbreviated before. Check throughout the text.
Methods;
9) I am not an expert in systematic reviews, but the method followed by the authors seems rigorous enough. However, throughout the method, I have detected typographical and grammatical errors that must be checked. Also, other minor details that in my opinion should be mentioned:
10) L78-79. 19 years was the maximum age of inclusion. What was the minimum? It must be detailed.
11) L80. Why did the authors choose the last 10 years range? It must be detailed.
Results and discussion
12) L121, 122 ... Please, a sentence must never begin with Arabic numbers.
13) L. 178 Something strange happens in this sentence… ”Seventeen studies [26–32] were”. Please, check.
14) L180. Authors wrote "average age of between 6 and 19 years". On the other hand, this is not an average, but a range. Please check your wording.
15) The discussion is well conducted. However, there are some points that should be strengthened. In the first place, it would be very useful for understanding if the authors started again by remembering the objective and locating the gap they intend to address.
16) In addition, sometimes, the authors limit themselves to repeating the results instead of delving into why they occur. More explanations supported by theory are necessary.
17)Finally, it would be really useful for the authors to add a section of practical implications of their results, which seem quite positive in relation to the benefits of YLGs on the health of children. At the same time, they should ask themselves what the limit is. I mean, while AVGs report benefits, there are also studies that show that the abusive use of screens and technological devices can have harmful effects on health. What is the threshold? This is a final point that should be addressed in detail.
Author Response
Dear Reviewer,
We appreciate all your comments. We thank you for the excellent comments and for the opportunity to revise the manuscript, improving its quality and bringing it within the standards required by IJERPH.
The abstract was adjusted and now includes the suggestion;
The introduction was re-written;
We have changed all the typos and grammatical mistakes throughout the manuscript. We have also added the criteria for inclusion according to age (e.g. children or adolescents between 3 and 19 years old). We decided to include investigations from the last 10 years, because systematic reviews on this subject started to be published at that time. Only one systematic review was published in 2006 (A descriptive epidemiology of screen-based media use in youth: a review and critique). However, it did not meet the inclusion criteria.
We have rearranged the discussion. We have added the aims to the new version of the manuscript and we have also touched on the gap we intended to address. Moreover, as per suggestion a section of practical implications has been included in the review.
Thanks

Round 2
Reviewer 2 Report
The authors should clearly indicate at the end of the discussion that in the review, given the short period of time to which refers, they have little bibliography to obtain strong conclusions and that similar studies will need to be carried out with more published works.
Author Response
Dear Reviewer, we thank you for the excellent comments.
We added this informations in limitations: this review, given the short period of time to which refers, have little bibliography to obtain strong conclusions and similar studies will need to be carried out with more published works.
Reviewer 4 Report
The authors have solved all of minor and major concerns. Congratulations.
Author Response
Dear Reviewer, we appreciate for the excellent comments, thanks!